# Journey through the Decades: The Evolution in Treatment and Shared Decision Making for Locally Advanced Rectal Cancer

**DOI:** 10.3390/cancers16162807

**Published:** 2024-08-09

**Authors:** Racquel S. Gaetani, Keren Ladin, Jonathan S. Abelson

**Affiliations:** 1Department of Colon and Rectal Surgery, Lahey Hospital and Medical Center, Burlington, MA 01805, USA; jonathan.s.abelson@lahey.org; 2Department of Community Health, Tufts University, Medford, MA 02155, USA

**Keywords:** colorectal cancer, locally advanced rectal cancer, nonoperative management, watch and wait, quality of life, shared decision making, decision aids

## Abstract

**Simple Summary:**

Rectal cancer is a disease that affects thousands of people each year. The treatment options for locally advanced rectal cancer have significantly improved and can involve a combination of surgery, chemotherapy, and radiation therapy. Recently, there has been increased adoption of a new approach called “watch and wait”, where eligible patients can avoid surgery. This article describes how treatment options for locally advanced rectal cancer have evolved, emphasizes the importance of involving patients in decision making, and introduces a new tool to help patients and doctors decide about treatment options for rectal cancer.

**Abstract:**

The management of locally advanced rectal cancer has undergone significant transformations over the decades and optimal treatment approaches continue to evolve. There have been numerous advances in surgery, chemotherapy, and radiation therapy from the first description of the abdominoperineal resection in 1908, timing of chemotherapy and radiation therapy in the late 20th and early 21st century, and most recently, the introduction of organ preservation or nonoperative management in 2004. Alongside these advancements, the concept of shared decision making in medicine has evolved, prompting a focus on patient-centered care. This evolution in practice has been fueled by a growing recognition of the importance of patient autonomy and the alignment of treatment options with patients’ values and preferences. With the growing number of possible treatment options, variability in patient counseling exists, highlighting the need for a standardized approach to shared decision making in locally advanced rectal cancer. This narrative review will describe the evolution of treatment options of locally advanced rectal cancer as well as the concept of shared decision making and decision aids, and will introduce a decision aid for patients with locally advanced rectal cancer who have achieved a complete clinical response and are eligible for watch and wait.

## 1. Introduction

Colorectal cancer is the second most common cause of cancer-related deaths for men and women in the United States. The American Cancer Society projects 46,220 new cases of rectal cancer in 2024, underscoring its rising prevalence [1,2]. The management of rectal cancer has evolved through the years from relying solely on surgery to incorporating adjuncts such as chemotherapy, radiation therapy (RT), and chemoradiotherapy in combination with surgery. Additionally, the discovery and utilization of imaging modalities such as magnetic resonance imaging (MRI), computed tomography (CT), high-definition endoscopy, and endorectal ultrasound have revolutionized the diagnostic accuracy of identifying rectal cancers as well as clinical staging. These advancements now allow for nonoperative management or watch-and-wait strategies for patients with rectal cancer.

The management of locally advanced rectal cancer (LARC) has evolved over the decades and remains a topic of debate. This manuscript will discuss the evolution of treatment of LARC and the landmark trials that guide treatment practices today (Figure 1). Furthermore, we will discuss the evolution of shared decision making and patient decision aids to support treatment decisions. Additionally, it will introduce a decision aid for patients with LARC who are eligible for watch and wait after a complete clinical response.

## 2. Early Approaches and Surgical Innovations

The surgical management of rectal cancer has undergone profound evolution since its inception. The abdominoperineal resection (APR) was introduced in 1908 and became the standard treatment of middle to lower rectal tumors until the low anterior resection (LAR) was introduced in the 1920s [3,4]. Despite these surgical advancements, rectal cancer remained a terminal diagnosis, with a recurrence rate approaching 100% [23]. A seminal point in rectal cancer surgery occurred with the introduction of total mesorectal excision (TME) by Dr. Bill Heald in 1982 [5]. This revolutionary technique involves the complete removal of the rectum along with the pararectal lymph nodes within the mesorectum via sharp dissection along the visceral pelvic fascia. Adoption of TME resulted in a significant reduction in recurrence rates while concurrently mitigating the risk of urinary and sexual dysfunction through the preservation of sacral nerves [24,25].

Prior to TME, the local recurrence rate and 5-year overall survival (OS) of LARC were approximately 30% and 45%, respectively, for conventional surgical techniques; this then improved to a less than 10% local recurrence rate and upwards of 75% 5-year OS with the adoption of TME [23,26,27,28]. Though TME has reduced the rates of postoperative complications, the procedure may still be associated with long-term sequalae. These symptoms including sexual and urinary dysfunction, fecal incontinence, and low anterior resection syndrome (LARS) due to inadvertent damage to pelvic autonomic nerves [29,30,31,32,33]. LARS is a condition characterized by a constellation of symptoms including fecal incontinence, frequent bowel movements, urgency, and clustering and has been shown to affect up to 70% of patients after TME [34]. Urinary dysfunction may include urinary incontinence or retention, with a higher risk associated with low anastomosis and excessive intraoperative blood loss [35]. Sexual dysfunction after TME can occur in both men and women, with men potentially experiencing erectile dysfunction and difficulty with ejaculation, and women potentially developing dyspareunia and vaginal dryness [36,37].

## 3. Introduction of Adjuvant and Neoadjuvant Therapies

### 3.1. Adjuvant Therapies

In the late 20th century, studies were conducted to determine the effects of adjuvant chemotherapy and/or adjuvant RT for the management of LARC (Table 1) [6,7,38]. These trials demonstrated that combination therapy with adjuvant chemotherapy and RT resulted in lower rates of recurrence, though there were no differences in OS and disease-free survival (DFS) [6,7,38].

The National Surgical Adjuvant Breast and Bowel Project (NSABP) study randomized patients with LARC to three treatment arms: surgery alone, surgery plus adjuvant chemotherapy or surgery plus adjuvant RT. The results demonstrated a significant improvement in DFS and OS with adjuvant chemotherapy (DFS: 53% vs. 30%, *p* = 0.006; OS: 65% vs. 43%, *p* = 0.05) and an improvement in DFS with adjuvant RT (45% vs. 30%, *p* = 0.05). Additionally, adjuvant chemotherapy significantly reduced local recurrence rates, though the *p* value was not reported [7]. The Gastrointestinal Tumor Study Group (GITSG) study randomized patients to either surgery only, adjuvant chemotherapy, adjuvant RT, or adjuvant chemoradiation therapy (CRT). This study found an improvement in DFS and locoregional recurrence with adjuvant treatment compared to surgery along (*p* = 0.05 and *p* = 0.0009, respectively) [6]. Additionally, Krook et al. demonstrates that CRT was able to reduce LARC local recurrence by 46% (*p* = 0.036; 95% CI 2 to 70) in addition to improving distant metastasis, DFS, and OS [38].

As a result of these trials, the National Institute of Health (NIH) published a consensus statement in 1990 concluding that adjuvant chemotherapy and RT improves local control and reduces recurrence and should be used in patients with LARC [39].

### 3.2. Neoadjuvant Radiation Therapy

Around the same time as the introduction of adjuvant therapies, simultaneous investigations were underway to identify the efficacy of preoperative RT (Table 2). These investigations were driven by concerns that the tumor bed might be less responsive to RT after surgery due to hypoxia of the tissue. As trials began investigating optimal neoadjuvant radiation treatments, two main schedules emerged: short-course RT (SCRT) administered in 25 Gy in 5 fractions over 1 week, and long-course chemoradiation therapy (LCCRT), administered in 40 to 50 Gy in 20 to 25 fractions over 4 to 5 weeks combined with a radiosensitizer, most commonly, concurrent 5-fluorouracil (5-FU)-based chemotherapy.

The Uppsala trial randomized patients with rectal cancer to receive either preoperative SCRT or postoperative RT (60 Gy in seven to eight weeks) and was the first to demonstrate that local recurrence rates were significantly lower in the neoadjuvant RT group (13% vs. 22%, *p* = 0.02) [40]. Subsequently, the Swedish Rectal Cancer Trial compared SCRT followed by surgery to surgery alone [8]. The study revealed a significant reduction in local recurrence at 5 years in the group that received neoadjuvant RT (11% versus 27%, *p* < 0.01), in addition to improved 5-year OS (HR 0.79, 95% CI 0.66 to 0.92) and 9-year cancer-specific survival (HR 0.69, 95% CI 0.55 to 0.83) in the neoadjuvant RT group [8]. However, a limitation of this study was that TME principles were not utilized.

Four years later, the Dutch Colorectal Cancer Group published a similar trial comparing SCRT followed by TME to TME alone. This trial found that neoadjuvant SCRT improved local recurrence at two years (2.4% in the RT plus TME group versus 5.3% in TME group, *p* < 0.01; HR 3.42, 95% CI 2.05 to 5.71), though it did not find a statistically significant difference in 2-year OS (HR 1.02, 95% CI 0.83 to 1.25, *p* = 0.84) [9]. In 2011, the Dutch Colorectal Cancer Group published 12-year follow-up data, which showed that the 10-year cumulative incidence of local recurrence was significantly lower in the neoadjuvant SCRT group (5% versus 11%, *p* < 0.01). However, no improvement in 10-year OS was observed [42]. Several other trials have demonstrated similar benefits of neoadjuvant SCRT [43,44,45,46,47,48,49,50].

The German CAO/ARO/AIO-94 trial, which aimed to compare outcomes of preoperative versus postoperative chemoradiotherapy for LARC, randomly assigned patients to receive either preoperative or postoperative chemoradiotherapy with 5-FU-based chemotherapy. The trial found that neoadjuvant chemoradiotherapy led to better compliance, local control, and sphincter preservation [10]. A 10-year follow-up of the trial showed improved 10-year local control in the preoperative group (HR 0.60, 95% CI 0.4 to 1.0, *p* = 0.048) but found no difference in the 10-year incidence of distant metastasis, OS, or DFS (distant metastasis: HR 0.98, 95% CI 0.76 to 1.28, *p* = 0.9; DFS: HR 0.92, 95% CI 0.72 to 1.19, *p* = 0.54; OS: HR 0.95, 95% CI 0.79 to 1.21, *p* = 0.85) [11]. Similarly, the European Organization for Research and Treatment of Cancer (EORTC) trial found that LCCRT improves local recurrence rates (*p* = 0.002) but did not have an effect on 5- or 10-year DFS and OS (DFS: HR 0.84, 95% CI 0.78 to 1.13; OS: HR 1.02, 95% CI 0.83 to 1.26) when compared to neoadjuvant RT [41].

The utilization of immunotherapeutic agents for patients with mismatch repair-deficient (dMMR) or microsatellite instability-high tumors as part of neoadjuvant therapy has gained significant traction, offering promising results in improving treatment outcomes and expanding the therapeutic options for patients with LARC. Antibodies targeting programmed cell death protein-1 (PD-1) or its ligand PD-L1 used as monotherapy, in conjunction with chemoradiotherapy, or TNT has demonstrated its ability to achieve high cCR rates, pCR rates and is well tolerated by patients [12,51,52,53,54].

A prospective study by Cercek et al. demonstrated a cCR rate of 100% (95% CI 74 to 100) after 6 months of anti-PD-1 monoclonal antibody therapy, therefore eliminating the need for chemoradiation therapy in these patients [51]. Additionally, the NECTAR multicenter prospective study evaluating the combination of PD-1 blockage with LCCRT found that this combination achieved a pCR of 40% (95% CI 27.6 to 53.8), indicating enhanced efficacy compared to historical data of chemoradiation therapy alone [12]. Furthermore, several investigations have evaluated the efficacy of PD-1 blockage with varying TNT regiments and have found a pCR rate of 32–56%. These studies collectively suggest that PD-1 blockage in combination with TNT can improve outcomes in patients with dMMR LARC [52,53,54]. Importantly, PD-1 blocking agents have favorable safety profiles with investigations observing relatively low rates of adverse events of grade 3 of higher, with nausea, dermatitis, and fatigue being the most observed toxicities. These findings suggest that PD-1 blocking agents may provide a promising alternative treatment regimen for those with dMMR LARC. This approach is well-tolerated, is associated with high rates of cCR and pCR rates and can help avoid the morbidity associated with chemoradiation therapy.

### 3.3. Total Neoadjuvant Therapy

Advancements in rectal cancer research during the 21st century were directed towards refining and exploring the optimal neoadjuvant chemoradiation therapies for LARC to improve patient adherence, local and distant recurrence rates and pathological complete response (pCR) rates. Findings from the German CAO/ARO/AIO-94 trial, and others, established neoadjuvant LCCRT followed by TME as the standard treatment approach for LARC, thus paving the way for the advent of total neoadjuvant therapy (TNT) (Table 3) [11,13,55,56,57,58,59,60,61,62,63,64,65]. TNT consists of chemotherapy either before or after RT in the neoadjuvant setting, followed by TME. Benefits of neoadjuvant chemotherapy include tumor downstaging, control of micro metastatic disease, and improved patient compliance to the treatment regimen especially as compared to postoperative chemotherapy. This regimen also has the potential to eliminate the gastrointestinal toxicities associated with chemotherapy while a patient has a stoma in place, minimizing stoma-related morbidity [56,66]. A retrospective cohort study found that the rate of pCR was significantly higher in patients receiving TNT than those receiving adjuvant chemotherapy (41% versus 27%) [62].

There are two landmark randomized control trials that investigate the efficacy of TNT in patients with LARC, the Rectal Cancer and Preoperative Induction Therapy Followed by Dedicated Operation (RAPIDO) trial and the Neoadjuvant Chemotherapy with FOLFIRINOX and Preoperative Chemotherapy for Patients with Locally Advanced Rectal Cancer (UNI-Cancer-PRODIGE-23) trial [14,15]. These investigations showed that TNT improved 3-year disease-related treatment failure (DRTF), 3-year DFS, and increased the rate of pCR compared to standard neoadjuvant chemoradiation thereby demonstrating the efficacy of the TNT approach [14,15]. The RAPIDO trial demonstrated an improvement in 3-year DFS (HR 0.68, 95% CI 0.49 to 0.97, *p* = 0.034), 3-year OS (HR 0.69, 95% CI 0.49 to 0.97, *p* = 0.034), distant metastasis rates (17% versus 25%), and pCR rates (28% versus 12%, *p* < 0.0001) with TNT compared to neoadjuvant chemoradiation therapy. A 5-year follow-up of the RAPIDO trial further confirmed the long-term benefits of TNT, showing higher rates of pCR and lower rates of distant metastasis [67]. The PRODIGE-23 trial demonstrated an improvement in 3-year DRTF (HR 0.75, 95% CI 0.60 to 0.95, *p* = 0.019) with no difference in OS (HR 0.92, CI 0.67 to 1.25, *p* = 0.59) with TNT. Several other trials have been conducted exploring the utilization of TNT, with similar results [63,68,69,70].

Despite the promising results of trials investigating TNT, the regimens in the trials and in practice vary in terms of radiation dose, radiation fractions, chemotherapy agents and sequence of neoadjuvant treatment modalities. The 2024 American Society of Colon and Rectal Surgeons (ASCRS) clinical practice guidelines recommend the use of TNT for LARC but does not provide specific guidance on the which treatment regimen to use, leaving the choice of schedule to the discretion of health care clinicians and institutions [16]. Current variation in practice includes SCRT versus LCCRT and induction chemotherapy (systemic chemotherapy followed by RT) versus consolidation chemotherapy (RT followed by systemic chemotherapy).

Regardless of treatment regimen, TNT has led to a reduction in local recurrence to 6-7% from approximately 12% in TME alone and is associated with a pCR of approximately 30%. TNT has also been shown to reduce distant metastasis and increase DFS and OS compared to prior neoadjuvant treatment regimens [9,11,62,71,72,73].

Investigations in the 21st century also focused on establishing the optimal chemotherapeutic agents for the treatment of rectal cancer. In the late 20th century, colon and rectal cancers were treated with a chemotherapy regimen of 5-FU, semustine, and vincristine, which had significant toxicity profiles. The National Surgical Adjuvant Breast and Bowel Project Protocol (NSABP) C-03 trial demonstrated that 5-FU with the addition of leucovorin significantly prolonged 3-year DFS and OS compared to the original chemotherapy regimen in colon cancer patients [74]. Additionally, the German CAO/ARO/AIO-04 trial investigating the addition of oxaliplatin to 5-FU-based neoadjuvant chemoradiation therapy demonstrated that the addition of oxaliplatin resulted in improved 3-year DFS without increasing toxicity side effects [75]. Subsequent trials then demonstrated the efficiency of augmenting 5-FU and LV with oxaliplatin, leading to the development of the FOLFOX therapy regimen (Folinic acid, fluorouracil, and oxaliplatin) in the early 2000s [76,77,78]. This regimen is now the foundation for systemic treatment for nearly all nonmetastatic curable colon and rectal cancers.

The optimal interval between TNT and surgery in LARC is another critical factor influencing treatment outcomes. The Stockholm III trial compared SCRT followed by immediate surgery, SCRT with delayed surgery (4–8 weeks), and LCCRT with delayed surgery, finding that delaying surgery does not affect local recurrence rates. Additionally, the study found that there was no difference in postoperative complication rates among the three groups in the trial but, a pooled analysis showed that patients treated with SCRT followed by delayed surgery had a significantly lower rate of perioperative complications compared to those with SCRT and immediate surgery [79]. In another study by Akgun et al., patients with LARC were randomized to TME within 8 weeks or after 8 weeks following TNT. This trial demonstrated a significantly higher pCR rate in the group with a longer interval, with the highest pCR rate observed between 10 and 11 weeks [80]. Several other trials have investigated the optimal timing of surgical intervention after neoadjuvant treatment, generally recommending surgery 8–12 weeks postcompletion of TNT for LARC [81,82]. The NCCN and ASCRS guidelines for the management of rectal cancer recommend a multidisciplinary evaluation to tailor the timing of surgery based on individual patient factors [16,17,83].

## 4. Organ Preservation

Given the increased rate of pCR using TNT, along with the potential complications associated with TME such as temporary or permanent stoma, urinary and sexual dysfunction, and LARS, alternative approaches aimed at organ preservation were investigated. Nonoperative management (NOM) in LARC was pioneered by Dr. Habr Gama in 2004 and was coined “watch and wait” (W&W) [18]. In her index investigation on W&W, patients with T2–T4 and/or N+ disease who were treated with LCCR and had a clinical complete response (cCR) were enrolled in “observation therapy” [18]. The study defined a cCR as having no abnormalities on proctoscopy, digital rectal exam or on imaging. 26.8% of patients in this investigation achieved a cCR with two patients in the NOM group having a recurrence that was treated with either trans anal full-thickness excision or salvage brachytherapy. Most importantly, the study found that there was higher 10-year OS and DFS in the NOM versus surgical group [18].

The utilization of W&W has gained global acceptance, with ASCRS and NCCN incorporating definitions of a cCR in their guidelines. Both recommend a multimodal approach to assessment of cCR with no palpable tumor on digital rectal examination, no visible pathology other than a flat scar on endoscopy, and no evidence of disease on cross-sectional imaging (Table 4 [16,84]. MRI has become the preferred imaging modality for the initial staging, re-staging, and surgical planning in rectal cancer patients. MRI allows for a comprehensive evaluation of the tumor and the surrounding soft tissues, allowing clinicians to determine the tumor’s relationship to the mesorectal fascia, involvement of adjacent organs and lymph node involvement [85,86]. The standard MRI sequence for evaluating rectal cancer includes T2 weighed images pre- and postcontrast. However, diffusion-weighted MRI (DWI) has shown promise in improving diagnostic accuracy. DWI is an MRI technique that uses differences in the extracellular movement of water protons to differentiate between tissues [87]. Studies have been able to demonstrate that incorporating DWI improves the sensitivity for predicting a pCR compared to standard MRI and therefore improving clinician’s confidence in identifying patients who are eligible for W&W [87,88]. However, the interpretation of DWI results requires expertise and a nuanced understanding of rectal cancer imaging.

Several trials conducted on NOM have shown that W&W is feasible and is associated with high OS that is comparable to those who undergo TME. Rates of local regrowth are approximately 25–35%, with most amenable to salvage curative surgery (Table 5) [19,89,90,91,92,93,94,95]. Studies have demonstrated that 10 years after treatment, patients who choose the W&W approach have a DFS rate of 81% to 93% and an OS rate of 91% to 97%. Furthermore, a meta-analysis comparing W&W patients to those who underwent TME and were found to have a pCR found that there was no significant difference in OS [18,90,91,96,97]. Despite the many studies reporting favorable oncological outcomes, there are some studies that demonstrated inferior DFS and OS as well as higher rates of distant metastasis in patients in W&W [94]. These discrepancies in the published literature highlight the need for further research with well-matched study groups, adequately powered sample sizes with sufficient long-term follow-up to further evaluate the oncologic safety of W&W.

**Table 4 cancers-16-02807-t004:** Criteria for assessing clinical response.

	Complete Clinical Response	Near-Complete Clinical Response	Incomplete Clinical Response
Digital Rectal Exam	Normal	Smooth induration or minor mucosal abnormalities	Palpable tumor
Endoscopy	Flat white scar Telangiectasia Absence of ulcers and mucosal nodularity	Small mucosa nodules/minor mucosal irregularities Superficial ulcerations Mild, persistent erythema of the scar	Visible tumor
MRI-T2W	Only a dark T2 signal AND No visible lymph nodes	Moderately dark T2 signal, some intermediate signal AND/OR Partial regression of lymph nodes	More intermediate than a dark T2 signal, no T2 scar AND/OR No regression of lymph nodes

Adapted from sources: [16,98,99,100].

The limitation of the many studies performed assessing the safety and feasibility of W&W is the variability in TNT practices; specifically, few trials have addressed the effect of TNT sequence on organ preservation until recently. The multicenter Organ Preservation of Rectal Adenocarcinoma (OPRA) trial randomized 324 LARC patients to either induction therapy with FOLFOX or CAPOX (capecitabine with oxaliplatin) followed by LCCRT or consolidation therapy with LCCRT followed by chemotherapy again with either FOLFOX or CAPOX [20]. Patients with a cCR or near complete clinical response (ncCR) were followed by W&W while those with an incomplete response were treated with TME. The study demonstrated a higher rate of 3-year organ preservation with consolidation chemotherapy. Nevertheless, the trial revealed that regardless of TNT strategy, 3-year DFS was similar for patients who entered W&W compared to historical controls who were treated with neoadjuvant LCCRT followed by TME and adjuvant chemotherapy [20], thus providing assurances that W&W is an oncologically safe approach.

As a result of these findings, the ASCRS and the National Comprehensive Cancer Network (NCCN) both included a recommendation for the W&W approach in those who have achieved a cCR after TNT in their most recent practice guidelines [16,17]. Both guidelines suggest consideration of this approach in highly selected patients by experienced multidisciplinary teams and emphasize the need for a rigorous surveillance protocol. Both clinical practice guidelines discuss the need for further high-quality prospective data to assess long-term outcomes.

## 5. Shared Decision Making

With the many possible preoperative, operative, and postoperative management options of LARC comes a myriad of decisions for clinicians, patients, and families. From choosing between TNT regimens and surgical options to deliberating the appropriateness of organ preservation or need for adjuvant therapies, the decision-making process requires careful consideration. Each decision that is made during the patient’s treatment journey carries significant tradeoffs between possible clinical outcomes and long-term quality of life. Because there is no one dominant choice from a clinical standpoint, patients and families should consider the possible benefits and drawbacks alongside their preferences and values.

Shared decision making (SDM) is an approach where patients, in partnership with their clinicians actively participate in the decision-making process where there are multiple clinically acceptable treatment options [101]. This approach honors the role of patient as expert regarding their preferences and experience of illness, and the clinician as expert in the implications of different treatment options and prognosis. The Agency for Healthcare Research and Quality (AHRQ) has created a five-step process for engaging in SDM called the SHARE approach [21]. The model includes seeking participation from patients, helping patients explore and compare treatment options, assessing patients’ values and preferences, reaching a decision with a patient, and evaluating patients’ decision. This process has been shown to improve patient satisfaction and adherence to treatment recommendations. SDM approaches have additionally been shown to improve patient knowledge, produce more realistic expectations about what care can achieve, reduce decisional conflict, and increase the proportion of people actively participating in healthcare decision making [21,102].

### 5.1. Seeking Patient Participation and Assessing Patients’ Values and Preferences

It is important to understand a patient’s preference for involvement in healthcare decisions. Studies have demonstrated that sex, age, race, level of education, household income, culture beliefs, values and other patient-related factors can influence preferences for participation in health care decisions [103,104,105]. A systematic review looking at patient preferences for treatment and decision making among patients with colorectal cancer found that most patients favor a passive role in the decision-making process. These preferences, however, may vary based on factors such as age, gender, level of education, and disease severity with older patients and those with more severe disease preferring a passive role in decision making. Despite varying levels of desired involvement among patients, most express that receiving comprehensive information about their disease and treatment options is important in order for them to feel involved in the decision-making process [106].

### 5.2. Helping a Patient Explore and Compare Treatment Options

The NCCN Clinical Practice Guidelines in Oncology indicate that while some studies have shown promising DFS and OS rates in the W&W approach, others have not been able to replicate these results [84]. In terms of local recurrence, the W&W approach has shown a higher rate of local recurrence compared to those who undergo TME but, most recurrences, though not all, are salvageable with curative intent if managed appropriately. These findings have engendered skepticism and uncertainty amongst clinicians regarding the oncologic safety of W&W. Communicating these complexities requires a nuanced understanding of patients’ preferences. By providing patients with detailed information about the potential outcomes, treatment efficacy and impacts on quality of life, colon and rectal surgeons may be able to better enable patients and their loved ones to make informed decisions that align with their values and preferences, fostering SDM.

During the COVID-19 pandemic, many uncertainties arose surrounding the treatment and management of patients with LARC. With the widespread impact of the pandemic, existing treatment protocols for LARC were disrupted owing to constraints on elective surgery and oncology care, and there was a lack of clear evidence-based guidelines during this period. This made advising patients more difficult while also making SDM more important as health care providers and patients needed to collaboratively navigate the complexities of treatment while considering the evolving circumstances of the pandemic.

The pandemic led to a significant rise in the utilization of TNT, SCRT, W&W, and led to an increase in time from diagnosis to surgery [107,108,109]. In response to these challenges, clinical guidelines were adapted to better care for patients with LARC during the pandemic. Recommendations included increasing the utilization of SCRT, consider W&W for patients with a cCR, deferring elective surgeries for 6–12 weeks after treatment, and to utilize TNT, which demonstrated high compliance and good oncological outcomes. These adaptations aimed to provide effective treatment while accommodating the unique constraints of the pandemic, emphasizing the critical role of SDM in optimizing patient care and outcomes [109,110,111]. Ultimately, these changes stemmed from attempting to limit the number, frequency and duration of time patients spend in a healthcare setting.

When helping patients consider treatment options, quality of life considerations should be central to the decision-making process as patients must weigh the options between surgery and W&W. As noted earlier, TME may be associated with long-term sequalae that may negatively impact quality of life in patients with LARC, including sexual and urinary dysfunction, fecal incontinence, and LARS [29,30,31,32,33,34]. A matched-controlled study of 41 patients showed that those in W&W reported better physical functioning, cognitive functioning, global health status and lower pain scores compared to those who underwent TME after TNT [112]. Those who underwent TME also had significantly more fecal incontinence and LARS, though patients in both groups reported major LARS symptoms. The rates of sexual dysfunction were similar between the two groups, with the W&W group having more mild urinary symptoms compared to those who underwent TME [112]. Despite the potential improvements in quality of life with the W&W approach, many patients decide to undergo surgery for a multitude of reasons, including the definitive pathologic assessment of the cancer to confirm there is no residual disease and remove any residual cancer that may have been present but not detected with standard surveillance regimens.

### 5.3. Reaching and Evaluating a Decision with Patients

The final steps of the SHARE framework of SDM involve reaching and evaluating a decision with a patient that is a safe and feasible option on an individualized basis. Colorectal surgeons within the United States, and internationally, have varying practice patterns regarding the management of LARC after cCR, with significant differences in adoptions of the W&W approach. Additionally, not all medical centers have robust multidisciplinary infrastructure, including advanced imaging, rigorous follow-up protocols and a collaborate team of nurse navigators, oncologists, radiologists, and surgeons to support a W&W program. The lack of national data on the utilization of NOM further complicates the landscape, as there is no comprehensive information on clinicians’ utilization of this approach and beliefs regarding risks and benefits as it pertains to W&W. Furthermore, W&W approaches are not standardized across institutions, leading to variability in patients’ treatment options ultimately impacting patient access and knowledge of the W&W approach.

## 6. Decision Aid

In many contexts where treatment options and prognoses vary, SDM may be challenging due to the complexity of medical information, varying levels of health literacy among patients, and the emotional burden associated with making a health care decision. Patients may struggle to fully understand the risks and benefits of their options, which can lead to misalignment between their preferences and the chosen medical intervention or result in patients deferring the decision entirely to the provider. Patient decision aids may mitigate this challenge by clarifying the decision and providing evidence-based information about risk, benefits and outcomes in an easily digestible format that allows the patient to identify what matters most to them. A decision aid can be in the form of pamphlets, videos, or web-based tools. Studies have shown that the use of decision aids improves patient knowledge and improves agreement between patients and physicians [113,114].

The International Patient Decision Aid Standards (IPDAS) collaborative established a set of rigorous guidelines for the content, development, effectiveness, and implementation of decision aids [22]. The framework includes criteria for providing balanced information about options, presenting risks and benefits in a clear manner, presenting outcomes in an unbiased way, incorporating methods for clarifying patients’ values, and structured guidance in communication.

Patient decision aids have been utilized in a myriad of medical specialties including oncology, cardiovascular medicine, anesthesia, and in the palliative care setting as patients are faced with making choices about life-sustaining treatments [115,116,117,118,119,120]. The use of decision aids in colon and rectal surgery and rectal cancer have been used sporadically in select clinical settings starting in the early 2000s [121]. Wu et al. utilized a decision aid for patients newly diagnosed with rectal cancer to assist with the decision between LAR versus APR for mid to low rectal cancer and demonstrated a 37.5% improvement in patient knowledge as well as a reduction in decisional conflict, therefore enhancing SDM. Additionally, 96% of patients in this study recommended decision aids to others [122]. Another study utilizing focus groups to determine patient attitudes towards a decision aid about adjuvant treatment options for rectal cancer determined that patient knowledge was increased; patients in this study felt the decision aid would make decision making easier [123].

### Development of a Patient Decision Aid for Patients with LARC Eligible for W&W

Future clinical approaches to SDM may increasingly utilize patient decision aids, particularly given the novelty of the watch-and-wait strategy for patients with LARC who have achieved a cCR. Decision aids can be invaluable for patient who are faced with the decision between watch-and-wait and surgical intervention and may help facilitate SDM by enhancing patient understanding and helping to align values and preferences with treatment decisions.

In 2023, Smets et al., in Belgium, published a decision aid for rectal cancer patients who have achieved a cCR after TNT. The decision aid is designed to assist with the decision to pursue surgery versus W&W. Qualitative interviews with former rectal cancer patients and clinicians were used to evaluate the decision aid. The feedback indicated that incorporating the decision aid into practice would add value to the decision making process [124]. A study exploring a decision aid within the United Stated is important as patient populations may vary substantially in terms of race, ethnicity, socioeconomic status, and access to healthcare.

In a multicenter collaboration with Lahey Hospital and Medical Center in Burlington Massachusetts, Washington University at St. Louis Missouri and Tufts University in Medford, Massachusetts, a patient decision aid was created for patients with LARC who have achieved a cCR and are eligible for W&W. The creation of the decision aid began with a draft created following a literature review, and in accordance with the Ottawa Decision Support Framework, SDM frameworks, and adhering to IPDAS guidelines [22,125]. The draft then underwent alpha testing with researchers and clinicians involved in the development process to test content, comprehensibility, and usability.

The decision aid begins by defining key terms such as rectal cancer and clinical complete response and provides a brief overview of the two treatment options (Table 6). It then offers a comprehensive overview of the treatment options beginning with surgery. The surgery section of the decision aid includes information about recovery, potential short-term and long-term complications, cancer recurrence rates, costs, and frequency of follow-up appointments. The next section provides a comprehensive overview of active surveillance, detailing the surveillance schedule and the procedures involved in the watch-and-wait approach such as flexible sigmoidoscopy, digital rectal exam, and imaging. Additionally, it discusses oncologic outcomes, the benefits and risks of choosing active surveillance, treatment options if recurrence occurs, and associated costs. An advantages and disadvantages table compares surgery to the watch-and-wait approach in a side-by-side format, summarizing all the information into a single, easily digestible page. Finally, there is a section where patients can identify factors that influence their decision, along with space for comments and reflections to help guide the patient in the decision-making process.

The decision aid is currently undergoing the validation process through qualitative interviews with patients with LARC and colon and rectal surgeons. A nonvalidated version is provided as Appendix A. In addition to gathering feedback on clarity, effectiveness, compressibility, feasibility, and usability of the decision aid for both patients and clinicians we are also gathering information on patient experience and provider practices with offering and providing W&W. The patient interview guide focuses on key themes including patient preferences, values, treatment outcomes, involvement in decision making, shared decision making, satisfaction with the decision-making process and understanding of their condition and treatment options. The provider interview guide focuses on communication styles and information provisions. By integrating insight from both patients and clinicians, we aim to refine the decision aid.

## 7. Conclusions

The management of locally advanced rectal cancer has evolved dramatically over the decades and now is most commonly treated with total neoadjuvant therapy with the possibility of nonoperative management if a patient develops a clinical complete response. The integration of W&W and SDM in healthcare represents a progressive step in the management and counseling of LARC, emphasizing personalized patient care. SDM, facilitated by patient decision aids, empowers patients to participate in decision making and helps to align treatment decisions with patients values and preferences. More research is needed to standardize how we offer W&W to patients in the United States and to demonstrate that a decision aid can improve compliance with W&W surveillance programs, improve shared decision making and reduce decisional regret.

## Figures and Tables

**Figure 1 cancers-16-02807-f001:**
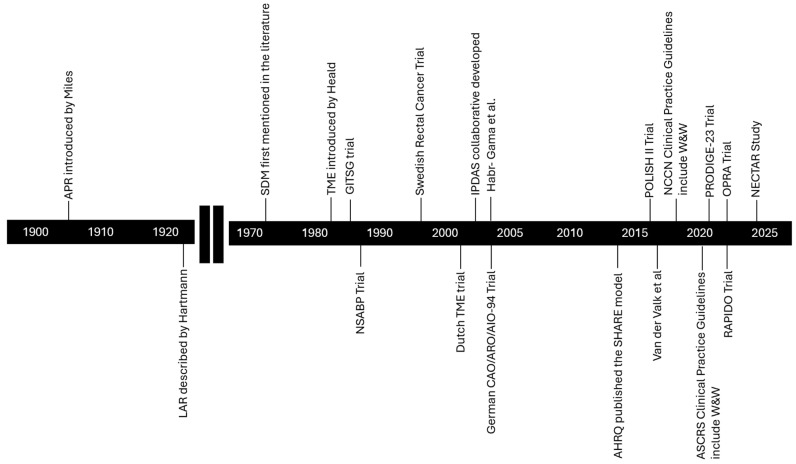
Timeline of events in the evolution of the treatment and management of LARC, SDM, and PtDAs [3,4,5,6,7,8,9,10,11,12,13,14,15,16,17,18,19,20,21,22].

**Table 1 cancers-16-02807-t001:** Key Clinical Trials Investigating Adjuvant Chemotherapy and/or Radiation Therapy for Locally Advanced Rectal Cancer.

Author/Study (Study Design)	Year	Rectal Cancers Included	n	Treatment Arms	5-Year Survival Outcomes	LRR	Toxicities	Findings
National Surgical Adjuvant Breast and Bowel Project (NSABP) Study [7] (RCT)	1988	Duke’s B and C	555	Surgery only	DFS: 30% OS: 43%	24.5%	Chemotherapy: hematologic (leukopenia, thrombocytopenia), GI (nausea, vomiting, mucositis) RT: diarrhea, proctitis, dermatitis, SBO, radiation enteritis	Improved 5-year DFS and OS when comparing the adjuvant chemotherapy group to adjuvant RT group. There was no benefit in 5-year DFS or OS with adjuvant RT
				Adjuvant chemotherapy (5-FU/semustine/vincristine)	DFS: 53% OS: 65%	21.4%		
				Adjuvant RT (46–47 Gy, 26–27 fx, 5 days per week; 51–53 Gy if boost)	DFS: 45% OS: 55%	16.3%		
Gastrointestinal Tumor Study Group (GITSG) [6] (RCT)	1985	Duke’s B_2_ and C	227	Surgery only	DFS: ~42% OS: ~44%	24%	Worse toxicities with CRT compared to chemotherapy or RT alone. Chemotherapy: leukopenia, nausea, vomiting, diarrhea RT: enteritis, diarrhea	Adjuvant CRT improves DFS and OS when compared to surgery alone
				Adjuvant chemotherapy (5-FU/semustine)	DFS: ~53% OS: ~57%	27%		
				Adjuvant RT (40 Gy in 4.5–5 weeks or 48 Gy in 5–5.5 weeks)	DFS: ~53% OS: ~51%	20%		
				Adjuvant CRT (40 Gy or 44 Gy in 4.5–5.5 weeks with 5-FU, followed by 5-FU/semustine	DFS: ~70% OS: ~60%	11%		
Krook et al. [38] (RCT)	1991	T3, T4, and/or N1 or N2	204	Adjuvant RT (45 Gy, 25 fx, 5 weeks; 5.4 Gy boost	DFS: ~37% OS: ~50%	25%	More toxicities with CRT compared to RT. Nausea, vomiting, diarrhea, leukopenia, and thrombocytopenia	Adjuvant CRT reduced DFS and OS, reduced relative recurrence by 34% and reduced length of time to recurrence
				Adjuvant CRT (45 Gy, 25 fx, 5 weeks; 5.4 Gy boost) with 5-FU → 5-FU/semustine	DFS: ~58% OS: ~58%	13.5%		

Abbreviations: CRT, chemoradiotherapy; DFS, disease-free survival; fx, fractions; 5-FU, 5-fluorouracil; GI, gastrointestinal; Gy, gray; LRR, locoregional recurrence rates; n, number of patients; OS, overall survival; RCT, randomized control trial; RT, radiation therapy; SBO, small bowel obstruction.

**Table 2 cancers-16-02807-t002:** Key Clinical Trials Investigating Neoadjuvant Chemotherapy and/or Radiation Therapy for Locally Advanced Rectal Cancer.

Author/Study (Study Design)	Year	Rectal Cancers Included	n	Treatment Arms	DFS	OS	LRR	Toxicities	Findings
Uppsala trial [40] (RCT)	1993	Duke’s B or C	471	Neoadjuvant SCRT (25.5 Gy in 1 week)	-	48%	13% †	SBO, ileus and proctitis though no difference was seen between groups	Neoadjuvant RT improves LRR but does not improve OS when compared to adjuvant RT
				Adjuvant RT (60 Gy in 7–8 weeks)	-	49%	22%		
Swedish Rectal Cancer Trial [8] (RCT)	1997	Duke’s A, B or C	1168	Surgery only	-	5-year 48%	5-year 27%	NR	Neoadjuvant SCRT improves LRR, DFS and OS compared to surgery alone
				Neoadjuvant SCRT (25 Gy, 5 fx, 1 week)	-	5-years 58% †	5-year 11% †		
Dutch TME Trial [9] (RCT)	2001	AJCC I-IV	1805	TME only	-	10-year 49%	10-year 11%	NR	Neoadjuvant SCRT improves LRR but does not improve OS compared to TME alone
				Neoadjuvant SCRT	-	10-year 48%	10-year 5% †		
German CAO/ARO/AIO-94 Trial [10,11] (RCT)	2004	T1 or T2 or T3 or T4 and/or N0/N+	824	Neoadjuvant LCCRT (50.4 Gy, 28 fx, 5 weeks) with 5-FU → TME	5-year 68% 10-year 68.1%	5-year 76% 10-year 59.6%	5-year 6% † 10-year 7.1% †	Fewer toxicities with neoadjuvant therapy. Diarrhea, hematological and dermatological effects	Neoadjuvant LCCRT improves LRR and has similar DFS and OS when compared to adjuvant LCCRT
				Adjuvant LCCRT (50.4 Gy, 28 fx, 5 weeks + boost 5.4 Gy) with 5-FU	5-year 65% 10-year 67.8%	5-year 74% 10-year 59.9%	5-year 13% 10-year 10.1%		
European Organization for the Research and Treatment of Cancer (EORTC) Trial [41] (RCT)	2006	T3, resectable T4M0 and/or N+	1011	Neoadjuvant RT (45 Gy, 25 fx, 5 weeks) → surgery	5-year 64.8% 10-year 50.7%	5-year – 10-year 50.7%	5-year 22% 10-year 22% †	Higher rate of toxicities with LCCRT compared to RT alone. Diarrhea, nausea, vomiting, neutropenia, radiation dermatitis	Neoadjuvant LCCRT improves LRR when compared to neoadjuvant RT Adjuvant chemotherapy with 5-FU/LV after neoadjuvant RT or LCCRT does not improve DFS or OS
				Neoadjuvant RT (45 Gy, 25 fx, 5 weeks) → surgery → adjuvant 5-FU/LV			5-year 13.7% 10-year 14.5%		
				Neoadjuvant LCCRT (45 Gy, 25 fx, 5 weeks) with 5-FU/LV → surgery			5-year 10.9% 10-year 11.8%		
				Neoadjuvant LCCRT (45 Gy, 25 fx, 5 weeks) with 5-FU/LV → surgery → adjuvant 5-FU/LV	5-year 65.8% 10-year 49.4%	5-year – 10-year 49.4%	5-year 10.7% 10-year 11.7%		

†: statistically significant, *p* value < 0.05 Abbreviations: AJCC, American Joint Committee on Cancer; DFS, disease-free survival; fx, fractions; 5-FU, 5-fluorouracil; Gy, gray; LCCRT, long-course chemoradiotherapy; LRR, locoregional recurrence rates; LV, leucovorin; n, number of patients; NR, not reported; OS, overall survival; RCT, randomized control trial; RT, radiation therapy; SCRT, short-course radiation therapy; TME, total mesorectal excision.

**Table 3 cancers-16-02807-t003:** Key Clinical Trials Investigating Total Neoadjuvant Therapy for Locally Advanced Rectal Cancer.

Authors/Study (Study Design)	Year	Rectal Cancers Included	n	Treatment Arms	Survival Outcomes	pCR	LRR	Toxicities	Findings
Grupo Cancer de Recto (GCR-3) Trial [58] (Phase II RCT)	2015	cT3, cT4 and/or cN+	108	Neoadjuvant LCCRT with oxaliplatin → TME → CAPOX	DFS: 5-year 64% OS: 5-year 78%	13.5%	5-year 2%	NR	Neoadjuvant CAPOX has similar DFS, OS, pCR, or LRR compared with adjuvant CAPOX
				Neoadjuvant CAPOX → neoadjuvant LCCRT with oxaliplatin → TME	DFS: 5-year 62% OS: 5-year 75%	14.3%	5-year 5%		
POLISH-II Trial [13] (Phase III RCT)	2016	Fixed cT3 or T4	541	Neoadjuvant RT (5 Gy for 5 days) → FOLFOX → TME	DFS: 3-year 53% OS: 3-year 73% †	16%	3-year 22%	Toxicities did not differ between the groups. Type of toxicities not specified.	Neoadjuvant RT followed by FOLFOX does not differ in DFS, OS, pCR, or LRR when compared to RT with simultaneous FOLFOX
				Neoadjuvant LCCRT (50.4 Gy, 28 fx) with FOLFOX → TME	DFS: 3-year 52% OS: 3-year 65%	11.5%	3-year 21%		
CAO/ARO/AIO-12 Trial [63] (Phase II RCT)	2019	cT3, cT4 and/or cN+	306	FOLFOX → CRT (50.4 Gy, 28 fx) with 5-FU and oxaliplatin) → TME	DFS: 3-year 73% OS: 3-year 92%	17%	6%	The group receiving chemotherapy first had higher rates of RT GI effects (diarrhea), hematologic, and neurologic toxicities when compared to the other group. Though this group had fever hematologic and neurotoxic effects of chemotherapy.	Consolidation chemotherapy results in higher pCR rates, no difference is seen in DFS, OS, or LRR between induction and consolidation chemotherapy TNT regimens
				CRT (50.4 Gy, 28 fx) with 5-FU and oxaliplatin) →FOLFOX → TME	DFS: 3-year 73% OS: 3-year 92%	25%	5%		
PRODIGE-23 Trial [14] (Phase III RCT)	2021	cT3 or cT4	461	Neoadjuvant CRT (50 Gy over 5 weeks) with capecitabine → TME → adjuvant FOLFOX or Capecitabine x8	DFS: 3-year 69% OS: 3-year 88%	12%	3-year 6%	The incidence of toxicities was similar between groups. Lymphopenia, neutropenia, neuropathy, diarrhea, nausea	Neoadjuvant FOLFIRINOX followed by CRT improved 3-year DFS and pCR rates compared to traditional CRT but did not improve OS or result in fewer LRRs
				Neoadjuvant FOLFIRINOX → CRT (50Gy over 5 weeks) with capecitabine → TME → adjuvant FOLFOX or Capecitabine	DFS: 3-year 76% † OS: 3-year: 91%	28% †	3-year 4%		
RAPIDO Trial [15] (Phase III RCT)	2021	cT4a/b, EMVI, cN2, involved MRF or enlarged LN	912	LCCRT (1.8–50.4 Gy, 28 fx or 2–50 Gy, 25 fx) with capecitabine → TME → optional adjuvant CAPOX or FOLFOX	DrTF: 3-year 30.4% OS: 3-year 89%	13.8%	3-year 6%	The incidence of toxicities was slightly higher in the TNT group. Diarrhea, neurological toxicity, neutropenia, lymphopenia	Neoadjuvant consolidation chemoradiotherapy improved 3-year DrTF and pCR compared to LCCRT + optional adjuvant chemotherapy
				Neoadjuvant RT (5 Gy for 5 days) → CAPOX or FOLFOX → TME	DrTF: 3 year 23.7% † OS: 3 year 89%	27.7% †	3-year 8.7%		

†: statistically significant, *p* value < 0.05. Abbreviations: CAPOX, Oxaliplatin and capecitabine; CRT, chemoradiotherapy; DFS, disease-free survival; DrTf, disease-related treatment failure; EMVI, extramural venous invasion; FOLFIRINOX, oxaliplatin and leucovorin followed by irinotecan and 5-fluorouracil; FOLFOX, 5-fluorouracil, leucovorin calcium (folinic acid), and oxaliplatin; fx, fractions; 5-FU, 5-fluorouracil; LCCRT, long-course chemoradiotherapy; LN, lymph node; LRR, locoregional recurrence rates; MRF, mesorectal fascia; n, number of patients; NR, not reported; OS, overall survival; pCR, pathological complete response; RT, radiation therapy; SCRT, short-course radiation therapy; TME, total mesorectal excision.

**Table 5 cancers-16-02807-t005:** Key Studies Investigating Watch-and-Wait Approach for Locally Advanced Rectal Cancer.

Authors/Study (Study Design)	Year	Rectal Cancers Included	n	Treatment Arms/Neoadjuvant Therapy Regimen	Survival Outcomes %	cCR	LRR	Findings
Habr Gama et al. [18] (Observational retrospective)	2004	cT1-4 N1-2	265	CRT (50.4 Gy/28 fx + 5-FU and leucovorin) → W&W in those with cCR	DFS: 5-year 92% OS: 5-years 100% †	27%	5-year 2.8%	There was a locoregional recurrence rate of 2.8% in the W&W group. There was no difference in DFS for those in W&W and those who had an iCR and underwent TME
				CRT → TME in those with iCR	DFS: 5-year 83% OS: 5-year 88%			
Habr Gama et al. [89] (Retrospective Cohort)	2014	cT2–cT4 or cN+	183	Neoadjuvant CRT (50.4–54 Gy) with 5-FU → assessed for tumor response 8 weeks after completion of RT	DFS: 5-year 68%	49%	31%	Salvage therapy possible in 93% of those with LR with a 5-year local recurrence-free survival rate of 94% and 5-year cancer-specific overall survival of 91%
Martens et al. [91] (Prospective Cohort)	2016	Rectal cancer without distant metastasis	100	CRT (1.8 Gy, 28 fx) with capecitabine or 5 Gy for 5 days → assessed for tumor response 8 weeks after completion of RT	DFS: 3-year 80.6% OS: 3-year 96.6%	61% nCR 39%	15%	W&W for cCR and nCR results in high 3-year OS and DFS
Van der Valk et al. [19] (International multicenter observational mixed prospective and retrospective)	2016	Rectal cancer who are entered into W&W	1009	Various—CRT most common (45 Gy, 50 Gy, 54 Gy or 60 Gy) with capecitabine or 5-FU	DFS: 5-year 94% OS: 5-year 84.7%		2-year 25.2%	Those in W&W had high 5-year OS and DFS 31% has local excision and 78% had salvage TME after recurrence
OPRA trial [20] (Prospective randomized phase II trial)	2022	Clinical stage II (T3-4, N0)—stage III (any T, N1-2)	324	Induction chemotherapy (FOLFOX or CAPOX) → CRT (4.5 Gy, 25 fx to nodes and 5–5.6 Gy to tumor) with capecitabine or 5-FU → NOM in cCR/nCR	DFS: 3-year 76% OS: 3-year ~ 95%	71% *	40%	Similar 3-year DFS were observed in those who underwent W&W compared to historical control and 3-year DFS did not differ amongst induction chemotherapy and consolidation chemotherapy. DFS was similar for those undergoing TME for iCR and for TME after re-growth
				CRT (4.5 Gy, 25 fx to nodes and 5–5.6 Gy to tumor) with capecitabine or 5-FU → consolidation chemotherapy (FOLFOX or CAPOX) → W&W in cCR/nCR	DFS: 3-year 76% OS: 3-year ~ 95%	76% *	27.5%	

†: statistically significant, *p* value < 0.05; * cCR and nCR; defined as complete endoluminal response to treatment (visible scar only) or residual scar/ulcer ≤3 cm in diameter. Abbreviations: CAPOX, capecitabine and oxaliplatin; cCR, complete clinical response; CRT, chemoradiotherapy; DFS, disease-free survival; FOLFOX, 5-fluorouracil, leucovorin calcium (folinic acid), and oxaliplatin; fx, fractions; 5-FU, 5-fluorouracil; iCR, incomplete clinical response; LCCRT, long-course chemoradiotherapy; LR, locoregional recurrence; LRR, locoregional recurrence rates; n, number of patients; nCR, near complete clinical response; OS, overall survival; pCR, pathological complete response; RT, radiation therapy; TAMIS, transanal minimally invasive surgery; TEM, transanal endoscopic microsurgery; TME, total mesorectal excision; W&W, watch and wait.

**Table 6 cancers-16-02807-t006:** Content of the patient decision aid for patients with LARC who have achieved a cCR and are eligible for watch and wait.

Section Title	Section Content
What is rectal cancer?	What is rectal cancer? What is a “clinical complete response”? How is clinical complete response in rectal cancer treated?
Surgery	Brief overview of the surgical options How long will I need to stay in the hospital after surgery? How long will my recovery time be? What are some possible short-term issues after surgery? Infection What are some possible long-term issues after surgery? Bowel function Stoma issues Sexual problems If I get surgery is there a risk of my cancer coming back? How much will surgery cost? How often will I need to see my doctor after surgery?
Active surveillance	What is active surveillance? How often will I have check-ups and what will they include if I choose active surveillance? What is the risk that my cancer will come back? What happens if my cancer comes back? What are the benefits to choosing active surveillance? Are there any risks to choosing active surveillance? How much does active surveillance cost?
Overview	Table depicting the options after clinical complete response
Advantages of surgery and active surveillance	Tabel of advantages and disadvantages
What factors affect your decision?	Table where patients can select whether certain factors affect their decision Comment section Area to write down “what are you most worried about?” Section to select “what are your next steps?”

## Data Availability

No new data was created or analyzed in this study. Data sharing is not applicable to this article.

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
