# Peer review of "Journey through the Decades: The Evolution in Treatment and Shared Decision Making for Locally Advanced Rectal Cancer"

_cancers, 2024, doi:10.3390/cancers16162807_

Round 1

Reviewer 1 Report

Comments and Suggestions for Authors

This is a well written manuscrpt that needs only few refinements as given below.

This is a well written manuscript on the evolution of treatment in rectal cancer.

I do have two points that need further refinement:

1.    RAPIDO-trial: A recent update oft he results from this trial revealed an excessive increase of locoregional failures (Dijkstra et al., 2023). Please include this detail in a footnote.

2.    Assessing response in patients with rectal cancer by MRI is a delicate procedure that needs a good expertise. Please refer to data given by Regina Beets-Tan particularly in terms of DWI imaging.

Author Response

July 29, 2024

Dear Editors,

We appreciate the reviewers’ time and thoughtful feedback on our manuscript. Revisions in the manuscript titled “Journey Through The Decades: Evolution in Treatment and Shared Decision Making for Locally Advanced Rectal Cancer” are highlighted within the manuscript. Below, we provide a detailed response to each comment and outline the changes made to address the suggested revisions.

Reviewer 1:

This is a well written manuscrpt that needs only few refinements as given below.

This is a well written manuscript on the evolution of treatment in rectal cancer.

 I do have two points that need further refinement:

Comments 1: RAPIDO-trial: A recent update of the results from this trial revealed an excessive increase of locoregional failures (Dijkstra et al., 2023). Please include this detail in a footnote.

Response 1: Thank you for your recommendation to include this reference. We agree with this comment and have added a sentence summarizing the findings of the 5-year follow-up of the RAPIDO trial. This addition can be found on Page 8, Paragraph 1, lines 189-196  Additionally, in response to reviewer 2’s comments we have also added a paragraph further describing the findings of the RAPIDO and PRODIGE-23 trials as noted below.

“The RAPIDO trial demonstrated an improvement in 3- year DFS (HR 0.68, 95% CI 0.49 to 0.97, p = 0.034), 3-year OS (HR 0.69, 95% CI 0.49 to 0.97, p = 0.034), distant metastasis rates (17% versus 25%), and pCR rates (28% versus 12%, p < 0.0001) with TNT com-pared to neoadjuvant chemoradiation therapy. A 5-year follow-up of the RAPIDO trial further confirmed the long-term benefits of TNT, showing higher rates of pCR and lower rates of distant metastasis [60]. The PRODIGE-23 trial demonstrated an im-provement in 3-year DRTF (HR 0.75, 95% CI 0.60 to 0.95, p = 0.019) with no difference in OS (HR 0.92, CI 0.67 to 1.25, p = 0.59) with TNT.”

Comments 2: Assessing response in patients with rectal cancer by MRI is a delicate procedure that needs a good expertise. Please refer to data given by Regina Beets-Tan particularly in terms of DWI imaging.

Response 2: We agree that it is important to highlight the role of MRI in assessing patient response to treatment. We have added a paragraph discussing the utilization of MRI, emphasizing how the inclusion of DWI can enhance the sensitivity and accuracy of detecting a complete clinical response in patients following total neoadjuvant therapy. This addition can be found on page 9, Paragraph 3, lines 225 – 271 as noted below:

“The utilization of W&W has gained global acceptance, with ASCRS and NCCN incorporating definitions of a cCR in their guidelines. Both recommend a multimodal approach to assessment of cCR with no palpable tumor on digital rectal examination, no visible pathology other than a flat scar on endoscopy, and no evidence of disease on cross-sectional imaging (Table 5) [64, 80]. MRI has become the preferred imaging mo-dality for the initial staging, re-staging, and surgical planning in rectal cancer patients. MRI allows for a comprehensive evaluation of the tumor and the surrounding soft tissues, allowing clinicians to determine the tumor’s relationship to the mesorectal fascia, involvement of adjacent organs and lymph node involvement [81, 82]. The standard MRI sequence for evaluating rectal cancer includes T2 weighed images pre- and post- contrast. However, diffusion-weighted MRI (DWI) has shown promise in improving diagnostic accuracy. DWI is an MRI technique that uses differences in the extracellular movement of water protons to differentiate between tissues [83]. Studies have been able to demonstrate that incorporating DWI improves the sensitivity for predicting a pCR compared to standard MRI and therefore improving clinician’s con-fidence in identifying patients who are eligible for W&W [83, 84]. However, the inter-pretation of DWI results requires expertise and a nuanced understanding of rectal cancer imaging.”

We sincerely thank you for your insightful review of our manuscript. We appreciate your constructive comments and recommendations which have been instrumental in refining our manuscript. Thank you for your time and effort in reviewing our work.

Reviewer 2 Report

Comments and Suggestions for Authors

Summary: This is an informative, well-structured review discussing the developments in locally advanced rectal cancer treatment and introduces a decision aid for patients eligible for watch-and-wait post-treatment. 

Comments: 

1. Kindly do not leave any information with a reference–for example "Sexual dysfunction after TME can occur in both men and women, with men potentially experiencing erectile dysfunction and difficulty with ejaculation, and  women potentially developing dyspareunia and vaginal dryness. "

2. Tables 1–3. To clarify the differences/similarities between the treatment arms, it is suggested to add the statistical results , including hazard ratio (or odds ratio) in addition to the 95% CI and P-value. Authors can add this information to the main text. 

3. Tables 1–3. Given the importance of patient's quality of life in selecting an appropriate treatment, it is suggested to add the information regarding the toxicity reports of the trials. 

4. The trials evaluating the immunotherpeutic agents are missed in this study. It is recommended to include these trials–for example NECTAR trial (https://www.nature.com/articles/s41392-024-01762-y), neoadjuvant anti–PD-1 in Mismatch Repair–Deficient LARC (https://www.nejm.org/doi/full/10.1056/NEJMoa2201445), among others. 

5. In the "Shared Decision Making" section it is suggested to add a subheading regarding the consensus on the management of LARC during COVID-19 pandemic–for example: 

https://link.springer.com/article/10.1007/s12029-020-00454-4

This information is valuable for readers in a similar situations in future. 

Author Response

July 29, 2024

Dear Editors,

We appreciate the reviewers’ time and thoughtful feedback on our manuscript. Revisions in the manuscript titled “Journey Through The Decades: Evolution in Treatment and Shared Decision Making for Locally Advanced Rectal Cancer” are highlighted within the manuscript. Below, we provide a detailed response to each comment and outline the changes made to address the suggested revisions.

Reviewer 2:

This is an informative, well-structured review discussing the developments in locally advanced rectal cancer treatment and introduces a decision aid for patients eligible for watch-and-wait post-treatment. 

Comments 1: Kindly do not leave any information with a reference–for example "Sexual dysfunction after TME can occur in both men and women, with men potentially experiencing erectile dysfunction and difficulty with ejaculation, and women potentially developing dyspareunia and vaginal dryness. "

Response 1: Thank you for your thorough review of our manuscript. We agree with your comment and have added a reference to the above sentence. This revision can be found on page 3, paragraph 1, line 83, as noted below.

“Sexual dysfunction after TME can occur in both men and women, with men potentially experiencing erectile dysfunction and difficulty with ejaculation, and women poten-tially developing dyspareunia and vaginal dryness [19, 20]”

Comments 2: Tables 1–3. To clarify the differences/similarities between the treatment arms, it is suggested to add the statistical results, including hazard ratio (or odds ratio) in addition to the 95% CI and P-value. Authors can add this information to the main text. 

Response 2: Thank you for your recommendation to include statistical results regarding the studies listed in Tables 1-3. We agree with your suggestion and believe that this addition enhances the comprehensiveness of the manuscript. To address this issue, we have included a paragraph detailing the statistical results for the studies presented in table 1. This can be found on page 3, paragraph 3, lines 91-103. Unfortunately, the National Surgical Adjuvant Breast and Bowel Project (NSABP) or the Gastrointestinal Tumor Study Group (GITSG) study did not report hazards ratio or odds ratio; therefore, results of this trial were reported in percentages accompanied by p-values. Additionally, the study by Krook et al. reported its findings as percent reduction, which we have included along with the corresponding 95% confidence intervals and p values for clarity. Please see below for the edits to the manuscript:

“The National Surgical Adjuvant Breast and Bowel Project (NSABP) study randomized patients with LARC to three treatment arms: surgery alone, surgery plus adjuvant chemotherapy or surgery plus adjuvant RT. The results demonstrated a sig-nificant improvement in DFS and OS with adjuvant chemotherapy (DFS: 53% vs. 30%, p = 0.006; OS: 65% vs. 43%, p = 0.05) and an improvement in DFS with adjuvant RT (45% vs. 30%, p = 0.05). Additionally, adjuvant chemotherapy significantly reduced local recurrence rates, though the p value was not reported [23]. The Gastrointestinal Tumor Study Group (GITSG) study randomized patients to either surgery only, adju-vant chemotherapy, adjuvant RT, or adjuvant chemoradiation therapy (CRT). This study found an improvement in DFS and locoregional recurrence with adjuvant treatment compared to surgery along (p = 0.05 and p = 0.0009 respectively) [21]. Addi-tionally, Krook et al demonstrates that CRT was able to reduce LARC local recurrence by 46% (p = 0.036; 95% CI 2 to 70) in addition to improving distant metastasis, DFS, and OS [22].”

Regarding Table 2, the associated hazards ratios, 95% confidence intervals, and p-values for the studies listed have been incorporated into the text of the manuscript. These revisions can be found throughout the text on pages 5 and 6 between lines 126-153. The Uppsala trial did not report hazards ratio therefore the results of his study were reported in percentage and p-value within the text. Please see updated Table 2 below:

Table 2. Key Clinical Trials Investigating Neoadjuvant Chemotherapy and/or Radiation Therapy for Locally Advanced Rectal Cancer.

Author/study

(study design)

Year

Rectal cancers included

N

Treatment arms

DFS

OS

LRR

Toxicities

Findings

Uppsala trial [25]

(RCT)

1993

Duke’s B or C

471

Neoadjuvant SCRT (25.5 Gy in 1 week)

-

48 %

13 %

SBO, ileus and proctitis though no difference was seen between groups

Neoadjuvant RT improves LRR but does not improve OS when compared to adjuvant RT

Adjuvant RT (60 Gy in 7-8 weeks)

-

49 %

22 %

Swedish Rectal Cancer Trial [26]

(RCT)

1997

Duke’s A, B or C

1,168

Surgery only

-

5-year 48 %

5-year 27 %

NR

Neoadjuvant SCRT improves LRR, DFS and OS compared to surgery alone

Neoadjuvant SCRT (25 Gy, 5 fx, 1 week)

-

5-years 58 %

5-year 11 %

Dutch TME Trial [27]

(RCT)

2001

AJCC I-IV

1,805

TME only

-

10-year 49 %

10-year 11 %

NR

Neoadjuvant SCRT improves LRR but does not improve OS compared to TME alone

Neoadjuvant SCRT

-

10-year 48 %

10-year 5%

German CAO/ARO/AIO-94 Trial [37,38]

(RCT)

2004

T1 or T2 or T3 or T4 and/or N0/N+

824

Neoadjuvant LCCRT (50.4 Gy, 28 fx, 5 weeks) with 5-FU → TME

5-year 68 %                 10-year 68.1 %

5-year 76 %

10-year 59.6 %

5-year 6 %

10-year 7.1 %

Fewer toxicities with neoadjuvant therapy.

Diarrhea, hematological and dermatological effects

Neoadjuvant LCCRT improves LRR and has similar DFS and OS when compared to adjuvant LCCRT

Adjuvant LCCRT (50.4 Gy, 28 fx, 5 weeks + boost 5.4 Gy) with 5-FU

5-year 65 %

10-year 67.8 %

5-year 74 %

10-year 59.9 %

5-year 13 %

10-year 10.1 %

European Organization for the Research and Treatment of Cancer (EORTC) Trial [39]

(RCT)

2006

T3, resectable T4M0 and/or N+

1,011

Neoadjuvant RT (45 Gy, 25 fx, 5 weeks) → surgery

5-year 64.8 %

10-year 50.7 %

5-year –

10-year 50.7 %

5-year 22 %

10-year 22 %

Higher rate of toxicities with LCCRT compared to RT alone.

Diarrhea, nausea, vomiting, neutropenia, radiation dermatitis

Neoadjuvant LCCRT improves LRR when compared to neoadjuvant RT

Adjuvant chemotherapy with 5-FU/LV after neoadjuvant RT or LCCRT does not improve DFS or OS

Neoadjuvant RT (45 Gy, 25 fx, 5 weeks) → surgery → adjuvant 5-FU/LV

5-year 13.7 %

10-year 14.5 %

Neoadjuvant LCCRT (45 Gy, 25 fx, 5 weeks) with 5-FU/LV → surgery

5-year 10.9 %

10-year 11.8 %

Neoadjuvant LCCRT (45 Gy, 25 fx, 5 weeks) with 5-FU/LV → surgery → adjuvant 5-FU/LV

5-year 65.8 %

10-year 49.4 %

5-year –

10-year 49.4 %

5-year 10.7 %

10-year 11.7 %

†: statistically significant, p value < 0.05 Abbreviations: AJCC, American Joint Committee on Cancer; DFS, disease-free survival; fx, fractions; 5-FU, 5-fluorouracil; Gy, gray; LCCRT, long-course chemoradiotherapy; LRR, locoregional recurrence rates; LV, leucovorin; NR, not reported; OS, overall survival; RCT, randomized control trial; RT, radiation therapy; SCRT, short course radiation therapy; TME, total mesorectal excision

Regarding Table 3, the only studies within this table that were specifically mentioned in the text were the RAPIDO trial and the PRODIGE-23 trial. The hazards ratios, 95% confidence intervals and p-values for these trials are detailed within the text on page 8, paragraph 1, lines 189-196. Although the table itself does not report specific statistical results, statistically significant results are represented with a †. We believe that adding the results of the additionally studies into the text would not substantially enhance the value of the paper, as the key findings and their significance are effectively communicated throughout the table and manuscript text.  Please see the updated Table 3 below:

Table 3. Key Clinical Trials Investigating Total Neoadjuvant Therapy for Locally Advanced Rectal Cancer.

Authors/study

(study design)

Year

Rectal cancers included

N

Treatment arms

Survival outcomes

pCR

LRR

Toxicities

Findings

Grupo Cancer de Recto (GCR-3) Trial [48]

(Phase II RCT)

2015

cT3, cT4 and/or cN+

108

Neoadjuvant LCCRT with oxaliplatin → TME → CAPOX

DFS: 5-year 64 %

OS: 5-year 78 %

13.5 %

5-year 2 %

NR

Neoadjuvant CAPOX has similar DFS, OS, pCR, or LRR compared with adjuvant CAPOX

Neoadjuvant CAPOX → neoadjuvant LCCRT with oxaliplatin → TME

DFS: 5-year 62 %

OS: 5-year 75 %

14.3 %

5-year 5 %

POLISH- II Trial [50]

(Phase III RCT)

2016

Fixed cT3 or T4

541

Neoadjuvant RT (5 Gy for 5 days) → FOLFOX → TME

DFS: 3-year 53 %

OS: 3-year 73 %

16 %

3-year 22 %

Toxicities did not differ between the groups. Type of toxicities not specified.

Neoadjuvant RT followed by FOLFOX does not differ in DFS, OS, pCR, or LRR when compared to RT with simultaneous FOLFOX

Neoadjuvant LCCRT (50.4 Gy, 28 fx) with FOLFOX → TME

DFS: 3-year 52 %

OS: 3-year 65 %

11.5 %

3-year 21 %

CAO/ARO/AIO-12 Trial [54]

(Phase II RCT)

2019

cT3, cT4 and/or cN+

306

FOLFOX → CRT (50.4 Gy, 28 fx) with 5-FU and oxaliplatin) → TME

DFS: 3-year 73 %

OS: 3-year 92 %

17 %

6 %

The group receiving chemotherapy first had higher rates of RT GI effects (diarrhea), hematologic, and neurologic toxicities when compared to the other group. Though this group had fever hematologic and neurotoxic effects of chemotherapy.

Consolidation chemotherapy results in higher pCR rates, no difference is seen in DFS, OS, or LRR between induction and consolidation chemotherapy TNT regimens

CRT (50.4 Gy, 28 fx) with 5-FU and oxaliplatin) →FOLFOX → TME

DFS: 3-year 73 %

OS: 3-year 92 %

25 %

5 %

PRODIGE-23 Trial [58]

 (Phase III RCT)

2021

cT3 or cT4

461

Neoadjuvant CRT (50Gy over 5 weeks) with capecitabine → TME → Adjuvant FOLFOX or   Capecitabine x8

DFS: 3-year 69 %

OS: 3-year 88 %

12 %

3-year 6 %

The incidence of toxicities was similar between groups.

Lymphopenia, neutropenia, neuropathy, diarrhea, nausea

Neoadjuvant FOLFIRINOX followed by CRT improved 3-year DFS and pCR rates compared to traditional CRT but did not improve OS or result in fewer LRRs

Neoadjuvant FOLFIRINOX → CRT (50Gy over 5 weeks) with capecitabine → TME → Adjuvant FOLFOX or Capecitabine

DFS: 3-year 76 %

OS: 3-year: 91%

28 %

3-year 4 %

RAPIDO Trial [59]

(Phase III RCT)

2021

cT4a/b, EMVI, cN2, involved MRF or enlarged LN

912

LCCRT (1.8 Gy – 50.4 Gy, 28 fx or 2 Gy – 50 Gy, 25 fx) with capecitabine → TME → optional adjuvant CAPOX or FOLFOX

DrTF: 3-year 30.4 %

OS: 3-year 89 %

13.8 %

3-year

6 %

The incidence of toxicities was slightly higher in the TNT group.

Diarrhea, neurological toxicity, neutropenia, lymphopenia

Neoadjuvant consolidation chemoradiotherapy improved 3-year DrTF and pCR compared to LCCRT + optional adjuvant chemotherapy

Neoadjuvant RT (5 Gy for 5 days) → CAPOX or FOLFOX → TME

DrTF: 3 year 23.7 %

OS: 3 year 89 %

27.7 %

3-year

8.7 %

: statistically significant, p value < 0.05 Abbreviations: CAPOX, Oxaliplatin and capecitabine; CRT, chemoradiotherapy; DFS, disease-free survival; DrTf, disease-related treatment failure; EMVI, extramural venous invasion; FOLFIRINOX, oxaliplatin and leucovorin followed by irinotecan and 5-fluorouracil; FOLFOX, 5-fluorouracil, leucovorin calcium (folinic acid), and oxaliplatin; fx, fractions; 5-FU, 5-fluorouracil; LCCRT, long course chemoradiotherapy; LN, lymph node; LRR, locoregional recurrence rates; MRF, mesorectal fascia; NR, not reported; OS, overall survival; pCR, pathological complete response; RT, radiation therapy; SCRT, short course radiation therapy; TME, total mesorectal excision

Comments 3: Tables 1–3. Given the importance of patient's quality of life in selecting an appropriate treatment, it is suggested to add the information regarding the toxicity reports of the trials. 

Response 3: Thank you for this recommendation. We also find this information important to include. Consequently, we have added a column for toxicity reports of the trials within Tables 1-3. These revisions can now be found within the updated tables as noted above. Updated table 1 below.

Table 1. Key Clinical Trials Investigating Adjuvant Chemotherapy and/or Radiation Therapy for Locally Advanced Rectal Cancer.

Author/study

(study design)

Year

Rectal cancers included

N

Treatment arms

5- year survival outcomes

LRR

Toxicities

Findings

National Surgical Adjuvant Breast and Bowel Project (NSABP) Study [23]

(RCT)

1988

Duke’s B and C

555

Surgery only

DFS: 30 %

OS: 43 %

24.5 %

Chemotherapy: hematologic (leukopenia, thrombocytopenia), GI (nausea, vomiting, mucositis)

RT: diarrhea, proctitis, dermatitis, SBO, radiation enteritis

Improved 5-year DFS and OS when comparing the adjuvant chemotherapy group to adjuvant RT group. There was no benefit in 5-year DFS or OS with adjuvant RT.

Adjuvant chemotherapy (5-FU/semustine/vincristine)

DFS:  53 %

OS:  65 %

21.4 %

Adjuvant RT (46-47 Gy, 26-27 fx, 5 days per week; 51-53 Gy if boost)

DFS:  45 %

OS:  55 %

16.3 %

Gastrointestinal Tumor Study Group (GITSG) [21]

(RCT)

1985

Duke’s B2 and C

227

Surgery only

DFS:  ~ 42 %

OS:  ~ 44 %

24 %

Worse toxicities with CRT compared to chemotherapy or RT alone.

Chemotherapy: leukopenia, nausea, vomiting, diarrhea

RT: enteritis, diarrhea

Adjuvant CRT improves DFS and OS when compared to surgery alone

Adjuvant chemotherapy (5-FU/semustine)

DFS:  ~ 53 %

OS:  ~ 57 %

27 %

Adjuvant RT (40 Gy in 4.5-5 weeks or 48 Gy in 5-5.5 weeks)

DFS:  ~ 53 %

OS:  ~ 51 %

20 %

Adjuvant CRT (40 Gy or 44 Gy in 4.5-5.5 weeks with 5-FU, followed by 5-FU/semustine

DFS:  ~ 70 %

OS:  ~ 60 %

11 %

Krook et al [22]

(RCT)

1991

T3, T4, and/or N1 or N2

204

Adjuvant RT (45 Gy, 25 fx, 5 weeks; 5.4 Gy boost

DFS: ~ 37 %

OS: ~ 50 %

25 %

More toxicities with CRT compared to RT. Nausea, vomiting, diarrhea, leukopenia, and thrombocytopenia

Adjuvant CRT reduced DFS and OS, reduced relative recurrence by 34% and reduced length of time to recurrence

Adjuvant CRT (45 Gy, 25 fx, 5 weeks; 5.4 Gy boost) with 5-FU → 5-FU/semustine

DFS: ~ 58 %

OS: ~ 58 %

13.5 %

Abbreviations: CRT, chemoradiotherapy; DFS, disease-free survival; fx, fractions; 5-FU, 5-fluorouracil; GI, gastrointestinal; Gy, gray;

LRR, locoregional recurrence rates; OS, overall survival; RCT, randomized control trial; RT, radiation therapy; SBO, small bowel obstruction

Comments 4: The trials evaluating the immunotherpeutic agents are missed in this study. It is recommended to include these trials–for example NECTAR trial (https://www.nature.com/articles/s41392-024-01762-y), neoadjuvant anti–PD-1 in Mismatch Repair–Deficient LARC (https://www.nejm.org/doi/full/10.1056/NEJMoa2201445), among others. 

Response 4: Thank you for highlighting this important aspect. We agree with your recommendation and think that this addition would enhance our manuscript. Therefore, we included a paragraph discussing the utilization of immunotherapeutic agents for patients with locally advanced rectal cancer. This may be found on page 6, paragraph 2, lines 155-161 as noted below:

The utilization of immunotherapeutic agents for patients with mismatch re-pair-deficient or microsatellite instability-high tumors as part of neoadjuvant therapy has gained significant traction, offering promising results in improving treatment outcomes and expanding the therapeutic options for patients with LARC. Antibodies targeting pro-grammed cell death protein-1 (PD-1) or its ligand PD-L1 used in conjunction with neoad-juvant chemotherapy and/or chemoradiation therapy has demonstrated its ability to achieve high cCR rates, pCR rates and is well tolerated by patients [40-44].

Comments 5: In the "Shared Decision Making" section it is suggested to add a subheading regarding the consensus on the management of LARC during COVID-19 pandemic–for example: 

https://link.springer.com/article/10.1007/s12029-020-00454-4

This information is valuable for readers in a similar situations in future. 

Response 5: We understand the importance of your suggestion and have added a discussion within the text to the shared decision-making section of our manuscript. We discuss changes in the management and treatment of patients with locally advanced rectal cancer during the COVID-19 pandemic, as well as recommendations for the management of this disease during the pandemic. These revisions may be found on page 13, paragraph 5, lines 359-361 and continuing to page 14, paragraph 1lines 362-375. We appreciate you highlighting this important aspect of patient care and shared decision-making. Please see below for the edits to the manuscript:

“During the COVID-19 pandemic, many uncertainties arose surrounding the treatment and management of patients with LARC. With the widespread impact of the pandemic, existing treatment protocols for LARC were disrupted owing to constraints on elective surgery and oncology care, and there was a lack of clear evidence-based guidelines during this period. This made advising patients more difficult while also making SDM more important as health care providers and patients needed to collaboratively navigate the complexities of treatment while considering the evolving circumstances of the pandemic.

The pandemic led to a significant rise in the utilization of TNT, SCRT, W&W, and led to an increase in time from diagnosis to surgery [103-105]. In response to these challenges, clinical guidelines were adapted to better care for patients with LARC during the pandemic. Recommendations included increasing the utilization of SCRT, consider W&W for patients with a cCR, deferring elective surgeries for 6-12 weeks after treatment, and to utilize TNT which demonstrated high compliance and good oncological outcomes. These adaptations aimed to provide effective treatment while accommodating the unique constraints of the pandemic, emphasizing the critical role of SDM in optimizing patient care and outcomes [105-107]. Ultimately these changes stemmed from attempting to limit the number, frequency and duration of time patients spend in a healthcare setting.”

We sincerely thank you for your thorough and insightful review of our manuscript. Your valuable feedback has greatly contributed to the quality and comprehensiveness of our manuscript. We thank you for your expertise and attention to detail and we are grateful for your input.

Reviewer 3 Report

Comments and Suggestions for Authors

The authors present a narrative review of the evolution of treatment of Rectal Cancer, additionally they present a view of the share decision making process between clinicians and patients. The manuscript is excelent, well structured and nicely written.

1. The objective of the works is clearly presented and it is pertinent. The treatment of rectal cancer has dramatically evolved during last decades and a miriad of chemo-radiotherapy programs have been introduced. In addition, there is no general consensus on the W&W schemes. The manuscript review the evolution of these therapies, with special interest in the W&W. Special interest is focused on the decision making process, with the patient as main protagonist in the decision.

2. The manuscript is well written, and presented, and easy to read. It is very informative.

3. The conclusions are consistent with the evidence presented and they address the main question posed.

6. The references are appropriate.

7. Tables are very clear and very informative  

Author Response

July 29, 2024

Dear Editors,

We appreciate the reviewers’ time and thoughtful feedback on our manuscript. Revisions in the manuscript titled “Journey Through The Decades: Evolution in Treatment and Shared Decision Making for Locally Advanced Rectal Cancer” are highlighted within the manuscript. Below, we provide a detailed response to each comment and outline the changes made to address the suggested revisions.

Reviewer 3:

The authors present a narrative review of the evolution of treatment of Rectal Cancer, additionally they present a view of the share decision making process between clinicians and patients. The manuscript is excelent, well structured and nicely written.

  1. The objective of the works is clearly presented and it is pertinent. The treatment of rectal cancer has dramatically evolved during last decades and a miriad of chemo-radiotherapy programs have been introduced. In addition, there is no general consensus on the W&W schemes. The manuscript review the evolution of these therapies, with special interest in the W&W. Special interest is focused on the decision making process, with the patient as main protagonist in the decision.
  2. The manuscript is well written, and presented, and easy to read. It is very informative.
  3. The conclusions are consistent with the evidence presented and they address the main question posed.
  4. The references are appropriate.
  5. Tables are very clear and very informative  

Response:

Thank you for your thorough review of our manuscript. We appreciate your positive feedback and thank you for your time and effort in reviewing our work.

Round 2

Reviewer 2 Report

Comments and Suggestions for Authors

Dear Authors

I appreciate your effort to revise the manuscript. All the comments are well-addressed in the revision, except for comment #4. Due to the importance of the immunotherapy in the rectal cancer management; it was expected the authors mention the details of the study, including the numerical extent of benefits, the statistical significance, and the possible toxicities. These pieces of information would enhance the content of the manuscript. 

Good Luck!

Author Response

August 2, 2024

Dear Editors,

Thank you for your additional comments. Below we have provided a detailed response to your comment and outline the changes made to address the suggested revisions.

Reviewer 2:

Comments 1:

Dear Authors

I appreciate your effort to revise the manuscript. All the comments are well-addressed in the revision, except for comment #4. Due to the importance of the immunotherapy in the rectal cancer management; it was expected the authors mention the details of the study, including the numerical extent of benefits, the statistical significance, and the possible toxicities. These pieces of information would enhance the content of the manuscript. 

Good Luck!

Response 1: Thank you for this suggested revision, we agree that the addition of further detail on the utilization of immunotherapy in locally advanced rectal cancer would enhance the content of the manuscript. We have provided more detail regarding efficacy as well as reported toxicities on page 6, paragraph 2, lines 158-172. This addition can be found below:

“A prospective study by Cercek et al demonstrated a cCR rate of 100% (95% CI  74 to 100) after 6 months of anti-PD-1 monoclonal antibody therapy, therefore eliminating the need for chemoradiation therapy in these patients [40]. Additionally, the NECTAR multicenter prospective study evaluating the combination of PD-1 blockage with LCCRT found that this combination achieved a pCR of 40% (95% CI 27.6 to 53.8), in-dicating enhanced efficacy compared to historical data of chemoradiation therapy alone [41].  Furthermore, several investigations have evaluated the efficacy of PD-1 blockage with varying TNT regiments and have found a pCR rate of 32-56%. These studies collectively suggest that PD-1 blockage in combination with TNT can improve outcomes in patients with dMMR LARC [42-44]. Importantly, PD-1 blocking agents have favorable safety profiles with investigations observing relatively low rates of adverse events of grade 3 of higher, with nausea, dermatitis, and fatigue being the most observed toxicities. These findings suggest that PD-1 blocking agents may provide a promising alternative treatment regimen for those with dMMR LARC. This approach is well-tolerated, is associated with high rates of cCR and pCR rates and can help avoid the morbidity associated with chemoradiation therapy.”

Again, we thank you for your time and valuable feedback on our manuscript.